# Inhibition of Pro-Inflammatory Cytokines by Metabolites of Streptomycetes—A Potential Alternative to Current Anti-Inflammatory Drugs?

**DOI:** 10.3390/microorganisms8050621

**Published:** 2020-04-25

**Authors:** Jiří Hrdý, Lenka Súkeníková, Petra Petrásková, Olga Novotná, David Kahoun, Miroslav Petříček, Alica Chroňáková, Kateřina Petříčková

**Affiliations:** 1Institute of Immunology and Microbiology, First Faculty of Medicine, Charles University, 116 36 Prague, Czech Republic; lenka.sukenikova@lf1.cuni.cz (L.S.); petra.petraskova@lf1.cuni.cz (P.P.); olga.novotna@lf1.cuni.cz (O.N.); miroslav.petricek@lf1.cuni.cz (M.P.); katerina.petrickova@lf1.cuni.cz (K.P.); 2Faculty of Science, University of South Bohemia, 370 05 České Budějovice, Czech Republic; david.kahoun@jcu.cz; 3Institute of Soil Biology, Biology Centre Academy of Sciences of the Czech Republic, 370 05 České Budějovice, Czech Republic; alica.chronakova@upb.cas.cz

**Keywords:** *Streptomyces*, secondary metabolites, manumycins, immunomodulation, inflammation

## Abstract

Current treatment of chronic diseases includes, among others, application of cytokines, monoclonal antibodies, cellular therapies, and immunostimulants. As all the underlying mechanisms of a particular diseases are not always fully clarified, treatment can be inefficient and associated with various, sometimes serious, side effects. Small secondary metabolites produced by various microbes represent an attractive alternative as future anti-inflammatory drug leads. Compared to current drugs, they are cheaper, can often be administered orally, but still can keep a high target-specificity. Some compounds produced by actinomycetes or fungi have already been used as immunomodulators—tacrolimus, sirolimus, and cyclosporine. This work documents strong anti-inflammatory features of another secondary metabolite of streptomycetes—manumycin-type polyketides. We compared the effect of four related compounds: manumycin A, manumycin B, asukamycin, and colabomycin E on activation and survival of human monocyte/macrophage cell line THP-1. The anti-cancer effect of manucycine A has been demonstrated; the immunomodulatory capacities of manumycin A are obvious when using micromolar concentrations. The application of all four compounds in 0.25–5 μM concentrations leads to efficient, concentration-dependent inhibition of IL-1β and TNF expression in THP-1 upon LPS stimulation, while the three latter compounds show a significantly lower pro-apoptotic effect than manumycin A. We have demonstrated the anti-inflammatory capacity of selected manumycin-type polyketides.

## 1. Introduction

Modern human society suffers from an alarming increase in immune system diseases associated with chronic inflammation. These vary in the target organ or organ system and also in severity of symptoms, ranging from mild disorders to serious life-threatening diseases [1]. Many of these, such as rheumatic arthritis, psoriasis, inflammatory bowel diseases, asthma, diabetes, etc., substantially decrease the quality of life and life-span and cause important economic losses. Inflammatory processes have recently been linked to many other diseases such as cancer [2,3] and atherosclerosis [4,5].

Though the effort to find a reliable treatment of inflammatory diseases is immense, breaking news on a universal drug with only mild side-effects are still missing. Currently, most of the clinically used, modern drugs are based on antibodies or agonists of inflammatory pathways components. Currently, these include antibodies directed to IL-1β (e.g., Anakinra) and TNF-α (e.g., Adalimumab), and IL-1R agonists (Rilonacept). Among novel therapeutic targets, JAK/STAT signaling pathways inhibitors were approved to treat rheumatic arthritis and inflammatory bowel diseases and are tested for various other inflammatory diseases. Low molecular weight inhibitors of JAK1 and JAK3 kinases such as Tofacitinib may be an example [6]. Different small compounds were found to target other signaling pathways: Enaminone E121 suppresses the TLR4 pathway [7] and TAK-242 binds to Toll-like receptors [8]. Immune cell recruitment represents another promising therapeutic target (cell adhesion block by, e.g., Vedolizumab in inflammatory bowel diseases), together with chemokine inhibition, such as CCX507 inhibitor of CCR9 [6].

Actinomycetes, soil filamentous bacteria, belong to well-known resources of bioactive compounds. Their secondary metabolism is immense and still substantially understudied. Two thirds of currently used antibiotics are based on the actinomycete secondary metabolites, but these bacteria also produce potent cancerostatics (daunorubicin, bleomycin, and actinomycin) and immunomodulators, e.g., FK506 and daptomycin [9]. Numerous actinomycete strains, often associated with marine invertebrates, also produce cancerostatics compounds [10]. Immunomodulatory features of their secondary metabolites are much less studied: FK506 and its derivatives are widely used in transplantation medicine. Other compounds are primarily used as antibiotics but also possess immunomodulatory features. Daptomycin combines antimicrobial activities with the suppression of an innate immune response by the neutralization of bacterial agonists of the innate immune response [11]. A strong anti-inflammatory effect has also been reported in several macrolide antibiotics—azithromycin, clarithromycin, erythromycin, and others [12,13,14]. However, the wider application of these drugs in the inflammatory diseases treatment is limited due to the fear of spreading the pathogen resistance to them, as they are still valuable as antibiotics mostly against both Gram-positive and Gram-negative bacteria. Our pilot experiments show that immunomodulatory features can often be identified in secondary metabolites extracts of many human-associated streptomycetes [15] and similar reports scarcely appear in other works [16,17,18,19].

The family of manumycin-related compounds is rather large, it includes several related molecules with the highest structural variability found in the upper polyketide chain. Only a few of them are produced by the producer strains in the amounts allowing purification of pure compounds in the amounts needed for detailed activity tests; most of them appear as side metabolites, or congeners of the main compound. This was one of the limits in the compound selection. Next, the selected compounds vary in the structure of their upper chains substantially—manumycins A and B have partially saturated, branched upper chains, asukamycin’s upper chain is unsaturated linear with a cyclohexyl attached in the end, and colabomycin E has a long, linear, unsaturated chain with an unusual *cis* isomery of a double bond (Figure 1). Manumycin-type antibiotics may be an excellent alternative to the macrolides. Their antibiotic effect is weak, and thus, they have never been used as antibacterials. They are short polyketides produced by various streptomycetes and related bacteria that exhibit strong pro-apoptotic and anti-inflammatory features. These activities originate in their enzyme-inhibitory features. The anti-cancer (pro-apoptotic) effect is based on the specific inhibition of Ras farnesylation by Ras-farnesyltransferase [20] and on the irreversible inhibition of cytosolic thioredoxin reductase TrxR-1 leading to overproduction of reactive oxygen species, ROS [21]. The anti-inflammatory features come from the inhibition of caspase 1, the enzyme needed for final processing of two crucial pro-inflammatory cytokines IL-1β and IL-18 [22,23,24] and from the inhibition of IKK kinase β subunit that breaks the function of the central regulator of immune response, NF-κB, and influences expression of the pro-inflammatory genes regulated by this factor [24,25]. It should not be neglected that manumycin A also inhibits neutral sphingomyelinase, which slows down the amyloid protein accumulation in the brain and subsequent neurodegeneration, as was shown in the mouse model [26].

As the best studied, and until recently the only commercially available, manumycin A has been assayed for its anti-cancer activity in numerous cancer cell lines [27,28,29]. However, its anti-inflammatory activities have long been neglected. We have shown that manumycin A downregulates the transcription of the following pro-inflammatory genes: *Il6*, *Tlr8*, *Il1b*, Il10, and *Egr1*, and inhibits IL-1β, IL-6, and IL-8 production in the TNF-α-stimulated THP-1 monocyte cell line and human peripheral blood monocytes in 0.25–1 mM concentrations. Additionally, IL-18 release was inhibited in THP-1 cells under these conditions. However, in concentrations over 2 mM, manumycin A negatively affected the cell viability in a dose-dependent manner, perhaps due to the induction of apoptosis [24].

The aim of this work was to assess and compare anti-inflammatory and cytotoxic activities of four metabolites from the manumycin family, the type compound, manumycin A [30], and three related compounds—newly commercially available smaller derivative manumycin B [31], asukamycin [32,33], and our newly discovered compound, colabomycin E [34]. The last two were isolated in our laboratory from the cultures of producer streptomycete strains. Based on our previous observation, cytokines IL-1β and TNF-α are the most sensitive markers reflecting downregulation of inflammation using THP-1 cell line in our experimental setup [15]. Therefore, the capacity of manumycin-type metabolites to suppress both gene expression and secretion of IL-1β and TNF-α has been tested. The particular enzyme-inhibitory activities are presumably driven by different metabolite’s molecule moieties. As they all together determine the overall effect of each compound on the human immune cells, we aimed to select a compound with the best anti-inflammatory, but weak pro-apoptotic features. Such compounds may represent an attractive candidate as a future anti-inflammatory drug lead.

## 2. Materials and Methods

### 2.1. Bacterial Strains and Active Compounds

Manumycin A was purchased from Sigma-Aldrich (St. Louis, MO, USA), manumycin B from Abcam (Cambridge, UK). Asukamycin A was isolated from the culture of *Streptomyces nodosus* ssp. *asukaensis* and colabomycin E from the culture of *Streptomyces aureus* SOK1/5-04 by a procedure described by Petrickova [34]. Strain SOK1/5-04 was isolated from colliery spoil heaps and is deposited in the Biology Centre Collection of Organisms (BCCO, No. BCCO 10_0005, www.actinomycetes.cz). The isolation quality check was performed using LC-MS described therein. All the compounds were dissolved in DMSO (Sigma-Aldrich (St. Louis, MO, USA), tissue culture grade) in 1 mM concentrations.

### 2.2. Cell Culture Conditions

Human monocytic leukemia cells THP-1 (purchased from American Type Culture Collection, ATCC) were used for the evaluation of the anti-inflammatory effect of manumycin-type metabolites. THP-1 cells were cultured as described previously [24]. Briefly, THP-1 cells were cultured in RPMI-1640 medium (Sigma–Aldrich, St. Louis, MO, USA) supplemented with 10% fetal calf serum, 2 mM L-glutamine, penicillin, and streptomycin (Sigma–Aldrich, St. Louis, MO, USA) in cell culture flasks passaged every third day. To evaluate the immunomodulatory effect of streptomycetes metabolites, cells were transferred to a 12 well plate and seeded at concentration 10^6^ cells/1 mL. The total volume of cell suspension was 2 mL/well. Cells were cultured under a 5.5% CO_2_ atmosphere at 37 °C. Cells were stimulated with LPS (1 μg/mL) together with manumycin-type metabolites (manumycin A, manumycin B, colabomycin E, and asukamycin) at three different concentrations (5 μM, 1 μM, and 0.25 μM). Only LPS-stimulated cells (1 μg/mL, cat. no. L-2654, Sigma Aldrich, St. Louis, MO, USA) were used as a positive control to evaluate the immunosuppressive capacity of manumycin-type metabolites. The impact of streptomycin on THP-1 viability was assessed by Trypan blue staining.

### 2.3. RNA Extraction

Total RNA was extracted as described previously [35]. Briefly, cells were transferred to 5 mL tubes, spun down, and the supernatant was discarded. Cell pellet was lysed, and RNA extracted using RNeasyMini kit (Qiagen, Hilden, Germany) according to the manufacturer recommendation. Concentration and purity of isolated RNA were checked by Nanodrop.

### 2.4. Quantitative RT-PCR

RNA was reverse transcribed as described previously [35] using a High Capacity cDNA Reverse Transcription Kit (4368813, ThermoFisher Scientific, Waltham, MA, USA). Gene expression of *Il1b* and *Tnfa* was determined using TaqMan Assays (*Il1b* Hs00174128_m1, *Tnfa* Hs00174128_m1). *Peptidyl prolyl isomerase A* (*Ppia*, Hs99999904_m1) was used as endogenous control. Relative quantification of gene expression was calculated as described previously [35]. Briefly, the Ct (number of the cycles needed to exceed the fluorescence threshold) of the housekeeping gene (*Pppia*) was subtracted from the Ct of the gene of interest (either *I11b* or *Tnfa*). This difference (ΔCt) was calculated for all samples. Next, ΔΔ Ct was calculated as the difference between ΔCt of the particular sample minus the average value of ΔCts of control samples (unstimulated THP1 cells). Finally, the results of gene expression were expressed as 2^-ΔΔCt^. 

### 2.5. Cytokine Detection by ELISA

The effect of manumycin-type metabolites on the capacity of THP-1 cells to release cytokines was measured by ELISA. After 24 h stimulation of THP-1 cells with manumycin-type metabolites, cell culture supernatants were collected and stored in -20 °C until processed. Concentration of cytokines was detected using kits for IL-1β (88-7261-88) and TNF-α (88-7346-88) both purchased from ThermoFisher Scientific (Waltham, MA, USA). Absorbance was read at 492 nm using ELISA reader (Tecan, Schoeller Instruments, Prague, Czech Republic). Concentration was subtracted from standard calibration curve using software KIM (Schoeller Instruments, Prague, Czech Republic).

### 2.6. Statistics

Results were statistically evaluated and graphically processed using Graphpad Prism (San Diego, CA, USA). Data were statistically analyzed using a t-test with a Bonferonni post-hoc test for multiple comparisons. Statistical significance was set at *p* ≤ 0.05. Results are expressed as a mean with the standard error from three independent experiments.

## 3. Results

### 3.1. The Effect of Manumycin-Type Metabolites on THP-1 Viability

We have determined the possible toxicity of novel streptomycetes metabolites (manumycin A—ManA; manumycin B—ManB; colabomycin E—Col; and asukamycin—Asu) on the viability of THP-1 cells at different time points. None of the metabolites had a substantial effect on the viability of the cell line in shorter exposure times (1 and 4 h) and in the assayed concentration range, Figure 2A,B. However, after 24 h, the cytotoxicity of ManA at 5 μM (*p* = 0.00097) and 1 μM (*p* = 0.0419) concentrations and ManB at a 5 μM concentration (*p* = 0.0279) was observed, Figure 2C. Both compounds isolated in our laboratory (Asu, Col) did not show the cytotoxic effects in the assayed concentration range.

### 3.2. The Capacity of Metabolites to Change the Gene Expression of Pro-Inflammatory Markers in THP-1

The anti-inflammatory effect of the model compound, manumycin A, has already been published [24]. The current study compared the anti-inflammatory effect of another three manumycin-type metabolites at different time points. The ability of ManA, ManB, Asu, and Col to influence the gene expression of pro-inflammatory markers (*Il1b, Tnfa*) was determined after 1 and 4 h of stimulation. Not surprisingly, LPS induced gene expression of *Il1b* (*p* = 0.0001). The highest concentration of ManA (5 μM) significantly suppressed gene expression of *Il1b* (*p* = 0.0002), whereas the lower 1 μM and 0.25 μM concentrations did not downregulate gene expression of LPS-induced *Il1b*expression in comparison with 5 μM ManA (*p* = 0.0001 and *p* = 0.0009, respectively) after 1 h of stimulation, Figure 3A.

Similarly, the 5 μM concentration of ManB downregulated LPS-induced expression of *Il1b* (*p* = 0.0002), but the gene expression was not suppressed at lower concentrations of ManB being significantly increased in comparison to 5 μM ManB (*p* = 0.0107 for 1 μM Man B and *p* = 0.0183 for 0.25 μM ManB). Asu limited LPS-induced gene expression of *Il1b*at concentration of 5 μM (*p* = 0.017), while 1 μM and 0.25 μM concentrations of Asu did not have any effect (*p* = 0.0151 and *p* = 0.0098, respectively). Lastly, the same effect was found in Col: *Il1b*expression was significantly suppressed only by the highest concentration of 5 μM (*p* = 0.0003). The other concentrations of Col used had no impact on suppression of *Il1b*expression in THP-1 cells (*p* = 0.011 and *p* = 0.012, respectively), Figure 3A.

Next, the level of *Tnfa* expression was followed as the second marker of THP-1 inflammatory response after 1 and 4 h of exposition to manumycin metabolites. Again, *Tnfa* gene expression was increased in LPS stimulated THP-1 cells (*p* = 0.0006). Similarly, as in the case of *Il1b*, all the compounds significantly lowered *Tnfa* expression in the highest applied concentration in comparion with THP-1 cells stimulated with LPS. Man A significantly downregulated the gene expression of *Tnfa* at the highest concentration (*p* = 0.0004) in comparison to the positive control (LPS stimulation); however, 1 μM and 0.25 μM concentrations of ManA did not suppress gene expression of *Tnfa* (*p* = 0.0043 and *p* = 0.0009, respectively). Similarly, the highest concentration of ManB limited TNF-α gene expression compared to LPS-stimulated cells positive control (*p* = 0.0004). Other tested concentration, did not affect gene expression in comparison with LPS stimulated cells (*p* = 0.0015 and *p* = 0.0021, respectively, for 1 μM and 0.25 μM ManB). A concentration of 5 μM Asu decreased gene expression of *Tnfa* after 1 h at the highest concentration (*p* = 0.0062), however, both lower concentrations had no effect on gene expression of *Tnfa* compared to LPS stimulated cells. A concentration of 5 μM Asu significantly limited gene expression of *Tnfa* in comparison to 1 μM and 0.25 μM concentrations of Asu (*p* = 0.005 and *p* = 0.0052 for 1 μM and 0.25 μM Asu, respectively). Finally, the highest concentration of Col significantly inhibited gene expression of *Tnfa* (*p* = 0.0008). The lower concentrations of Col had no impact on the *Tnfa* gene expression in comparison to LPS stimulated cells and gene expression of *Tnfa* was higher in cells stimulated with 1 μM Col and 0.25 μM Col compared to 5 μM Col (*p* = 0.0038 and *p* = 0.0007, respectively), see Figure 3B.

In order to get the pilot data on the kinetics of manumycin metabolite inhibitory action, the *Il1b* and *Tnfa* genes transcription profiles were assayed also after 4h of exposition to manumycin-type compounds. Indeed, the presence of LPS significantly promoted gene expression of *Il1b* (*p* = 0.0013). Similar to the shorter incubation time, 5 μM ManA substantially suppressed gene expression of *Il1b*in comparison to LPS stimulated cells (*p* = 0.0013) and 1 μM and 0.25 μM concentrations of ManA did not affect *Il1b*gene expression in comparison to LPS-stimulated cells but were higher compared to 5μM ManA used (*p* = 0.0011 and *p* = 0.0021, respectively). A concentration of 5 μM ManB lowered gene expression of *Il1b*in comparison to the LPS-stimulated sample (*p* = 0.0014). The other concentrations of ManB had no significant effect on LPS induced gene expression of *Il1b*. Gene expression of IL-1β in cells exposed to 0.25 μM ManB was significantly increased compared to 5 μM ManB (*p* = 0.0498). None of the Asu concentrations affected the gene expression of *Il1b*; however, 5 μM Col limited the gene expression of *Il1b*compared to LPS-stimulated cells (*p* = 0.0035). A concentration of 0.25 μM Col was less suppressive in comparison to 5 μM Col (*p* = 0.0323) (Figure 3C).

Correspondingly, the *Tnfa* gene expression was elevated in LPS-stimulated sample after 4 h incubation (*p* = 0.0084). A concentration of 5 μM ManA suppressed gene expression of *Tnfa* compared to the stimulated cells (*p* = 0.008). Gene expression of *Tnfa* was substantially higher in the samples affected by 1 μM ManA and 0.25 μM ManA in comparison to 5 μM ManA (*p* = 0.0048 and *p* = 0.0489, respectively). Similarly, 5 μM ManB inhibited gene expression of *Tnfa* in comparison to LPS-stimulated cells (*p* = 0.0083). Gene expression of *Tnfa* was different only in the 1 μM ManB stimulated sample compared to 5 μM ManB (*p* = 0.0499). A concentration of 5 μM Asu inhibited gene expression of *Tnfa* in comparison to LPS stimulated cells (*p* = 0.0496). Due to the high individual variability among particular experiments, other concentrations of Asu used had no impact on *Tnfa* expression in comparison with either LPS stimulated or 5 μM Asu stimulated THP-1 cells. Lastly, 5 μM Col inhibited gene expression of *Tnfa* after 4 h of incubation in comparison to LPS stimulated sample (*p* = 0.0266) (Figure 3D). The results of gene expression documented the concentration-dependent suppressive effect of manumycin-type metabolites on expression of pro-inflammatory markers in LPS-stimulated THP-1 cells. Interestingly, the effect of all the metabolites exhibited similar dynamics.

### 3.3. Effect of Manumycin-Type Metabolites on Cytokine Production

The suppressing effect of manumycin-type metabolites on the secretion of pro-inflammatory cytokines was measured by ELISA in cell culture supernatants after 24 h incubation. The concentration of IL-1β was increased in LPS stimulated sample compared to non-stimulated control (*p* = 0.0022). A profound effect of ManA on the decrease of IL-1β production was observed only when 5 μM ManA was used (*p* = 0.0028). Concentrations of 1 μM and 0.25 μM ManA were not able to lower IL-1β secretion in comparison to LPS-stimulated cells being still increased in comparison to 5 μM ManA (*p* = 0.0191 and *p* = 0.0482, respectively). A concentration of 5 μM ManB decreased secretion of IL-1β (*p* = 0.0027); however, 1 μM and 0.25 μM ManB were not able to inhibit IL-1β secretion as efficiently as 5 μM ManB (*p* = 0.0484 and *p* = 0.0485, respectively). Only a marginal suppressing effect on the pro-inflammatory marker IL-1β secretion by THP-1 cells supplemented with various concentrations of Asu was observed, but the highest concentration of Col limited release of IL-1β (*p* = 0.0358), Figure 4A.

The second pro-inflammatory marker production, TNF-α, was significantly elevated in the cell culture supernatants of THP-1 cells stimulated with LPS (*p* = 0.0411). Only 5 μM ManA attenuated TNF-α secretion in comparison to LPS stimulated sample (0.0441). Similarly, decrease of TNF-α was detected in cell culture supernatants of cells supplemented with 5 μM ManB (*p* = 0.0389). Concentrations of 1 μM and 0.25 μM ManB had no significant effect on the downregulation of TNF-α production. The concentration of TNF-α in cell culture supernatants of THP1 cells stimulated with LPS together with 0.25 μM ManB was significantly elevated in comparison with 5 μM ManB (*p* = 0.0102). A concentration of 5 μM Asu decreased production of TNF-α, but this did not reach statistical significance. Similarly, 1 μM and 0.25 μM Asu did not change production of TNF-α in comparison to 5 μM Asu or LPS. Only 5 μM Col inhibited secretion of TNF-α from LPS stimulated cells (*p* = 0.0497), Figure 4B. Concentration of pro-inflammatory markers in cell culture supernatants confirmed the data from gene expression analyses documenting the most profound effect of the highest tested concentration of manumycin-type metabolites on suppression of IL-1β and TNF-α release. The most effective metabolites are ManA, ManB, and Col. Based on lower toxicity of Col (Figure 2C) in comparison to ManA, Col seems to be a good candidate to be further tested for its anti-inflammatory properties.

## 4. Discussion

Manumycin-type metabolites show quite complex inhibitory effects on key human enzymes [20,21,24,25,26,36]. Many of these can be directly linked either to the regulation of cell proliferation and apoptosis or to the control of inflammatory response. Our previous study has shown that the most studied representative of this family of streptomycete secondary metabolites, manumycin A, exhibits both activities in similar active concentration in *in vitro* experiments using both human THP-1 monocyte/macrophage cell line and human peripheral blood monocytes. The pro-apoptotic and anti-inflammatory activities can be measured in as low as 1 μM concentrations of ManA and are concentration-dependent. This makes Man A an excellent candidate for a cancerostatic agent, which has been proven by numerous published studies before, e.g., the work of Di Paolo [28]. Additionally, it has been shown to work synergistically in combination with another cancerostatic agents, such as paclitaxel [37]. However, manumycin A—due to the high pro-apoptotic activity—cannot be considered a good candidate for a potential anti-inflammatory drug.

Our present study documents that other manumycin-type compounds show more suitable activity profiles. The cytotoxicity of all three other manumycins (ManB, Asu and Col) is substantially reduced, though their anti-inflammatory features remain the same or just slightly lower than in ManA. Though the set of structurally-related compounds is too small for a SAR (structure-activity relationship) assay, we can already relate some substructure-related activity changes. As the assayed compounds vary most exclusively in the structure of the upper polyketide chain, we expect that this part of the molecules is responsible for their variable cytotoxicity or pro-apoptotic features. The cytotoxicity should be, based on the current knowledge, directed by inhibition of two enzymes: the Ras-specific farnesyltransferase [20] and the TrxR-1cytosolic thioredoxin reductase [21]. The first uncouples the Ras G-protein from the cytosolic membrane and breaks up its signaling pathway. It can be speculated that the farnesyl transferase active site may bind the upper chain of manumycins due to its similarity to the farnesyl pyrophosphate, the donor of the farnesyl residue, as it is seen in some other FT inhibitors [38]. The strength of the competitive enzyme inhibition then depends on the structure of the upper chain. The second, TrxR-1-directed, inhibitory activity leads to an increased production of reactive oxygen species, one of the main elicitors of apoptosis [21,39], and is based on an irreversible adduct formation followed by conformational and activity changes. Similar mechanism has been shown also in algal brevetoxins [39]. This activity is supposed to rely on the central cyclic *m*C_7_N unit of manumycins; thus, we could expect that it will be less dependent on the structure of the variable upper polyketide chain and thus may be responsible for the basal pro-apoptotic activity found in all assayed manumycins. The same central part of the molecule is supposed to cause a covalent homodimerization of IKKβ leading to the inhibition of constitutive and TNF-α induced NF-κB activity with anti-inflammatory effects. The molecular mechanism of the second inflammatory response-inhibiting activity, the caspase 1 inhibition, has not been described yet.

In conclusion, our data suggest that manumycins may serve as promising drug for the suppression of excessive inflammation. Though the type molecule, manumycin A, shows too strong cytotoxicity, variations in its upper polyketide chain can change the ratio between pro-apoptotic and anti-inflammatory activities towards the second one. From this point of view, colabomycin E seems to be a better candidate for more detailed and mechanistic analysis of its anti-inflammatory potential. In general, the physicochemical properties of manumycins resemble those of plasma membrane phospholipids, it is generally expected that the compounds can cross the cytoplasmic membranes without any requirements for a transport system. The mouse model experiments also suggest their low toxicity and side-effects even in a long-time treatment [40]. We believe that streptomycetes and related actinomycetes produce even more immune cells-targeting active compounds. Our pilot experiments with selected, mostly human body-associated, streptomycetes showed that numerous strains produce various secondary metabolites with strong immunomodulatory effects (data not shown). This makes us believe that the immunomodulatory activities of streptomycete secondary metabolites have long been understudied and underestimated, compared to antibiotic, antifungal, and anti-cancer drugs. They should be considered as a valuable source for novel immunomodulatory drug leads discovery.

## Figures and Tables

**Figure 1 microorganisms-08-00621-f001:**
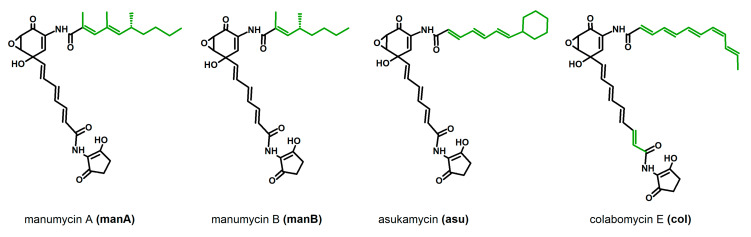
Structures of manumycin-type metabolites used in the study. Variable parts of molecules are shown in green. Manumycin A and B are produced by *Streptomyces parvulus* Tü64, asukamycin by *Streptomyces nodosus* ssp. *asukaensis,* and colabomycin E by *Streptomyces aureus* SOK1/5-04.

**Figure 2 microorganisms-08-00621-f002:**
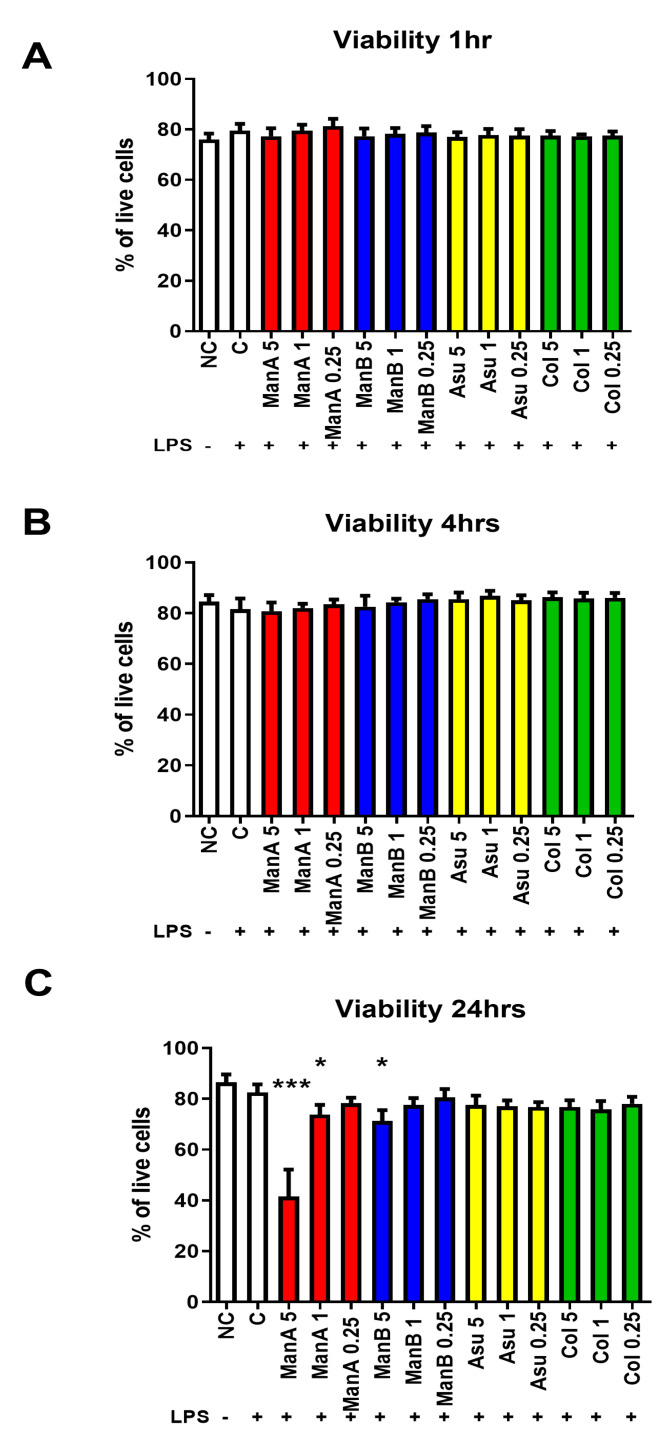
The impact of different manumycin-type metabolites on the viability of THP-1 cells. To assess the possible toxicity of manumycin-type metabolites, the viability of THP-1 cells was tested by Trypan blue using light microscopy at different time points. (**A**) viability of THP-1 cells after 1 h stimulation; (**B**) viability of THP-1 cells after 4 h stimulation; (**C**) viability of THP-1 cells after 24 h stimulation. Statistical significance values indicated as follows: * *p* ≤ 0.05, *** *p* ≤ 0.001. Sample legend: NC—non-stimulated control, LPS—lipopolysaccharide, ManA—manumycin A, ManB – manumycin B, Col—colabomycin E, Asu—asukamycin.

**Figure 3 microorganisms-08-00621-f003:**
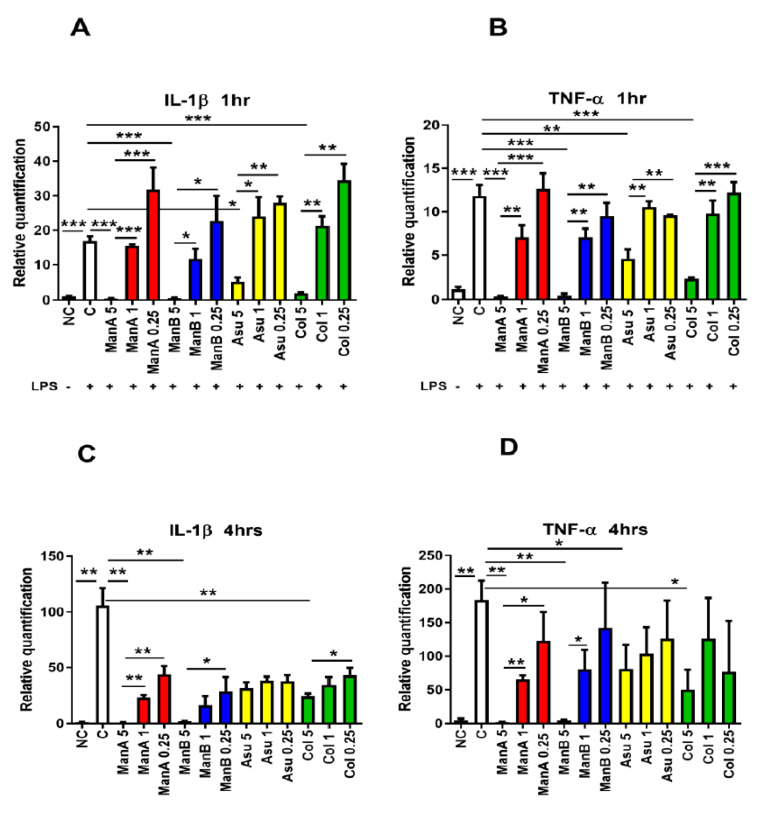
The effect of manumycin-type metabolites on gene expression of pro-inflammatory markers in THP-1 cells. THP-1 cells were stimulated with LPS and the effect of different concentrations of distinct manumycin-type metabolites on downregulation of expression of *Il1b*and *Tnfa* was tested by quantitative real-time PCR after 1 and 4 h stimulation. (**A**) the impact of manumycin-type metabolites on gene expression of *Il1b* in THP-1 cells after 1 h stimulation; (**B**) the impact of manumycin-type metabolites on gene expression of *Tnfa* in THP-1 cells after 1 h stimulation; (**C**) the impact of manumycin-type metabolites on gene expression of *Il1b* in THP-1 cells after 4 h stimulation; (**D**) the impact of manumycin-type metabolites on gene expression of *Tnfa* in THP-1 cells after 4 h stimulation; Statistical significance values indicated as follows: * *p* ≤ 0.05, ** *p* ≤ 0.01, *** *p* ≤ 0.001. Sample legend: NC—non-stimulated control, LPS—lipopolysaccharide, ManA—manumycin A, ManB – manumycin B, Col—colabomycin E, Asu—asukamycin.

**Figure 4 microorganisms-08-00621-f004:**
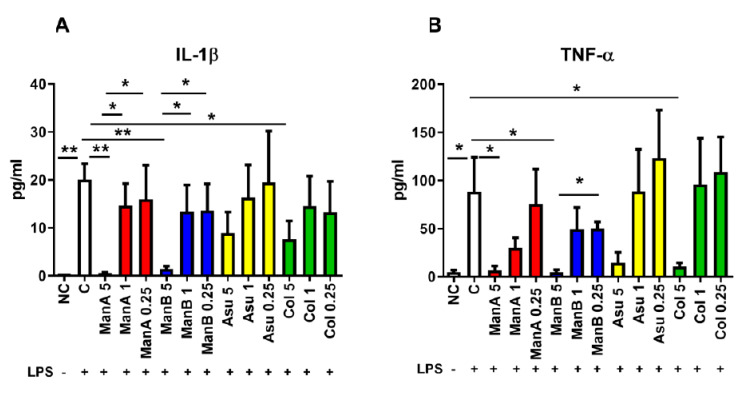
The impact of manumycin-type metabolites on secretion of pro-inflammatory markers by THP-1 cells has been tested by ELISA. The effect of various concentrations of manumycin-type metabolites on the capacity of THP-1 cells to release the pro-inflammatory cytokines IL-1β and TNF-α was tested by ELISA after 24 h stimulation. (**A**) IL-1β concentration in cell culture supernatants of THP-1 cells stimulated by manumycin-type metabolites; (**B**) TNF-α concentration in cell culture supernatants of THP-1 cells stimulated by manumycin-type metabolites; Statistical significance values indicated as follows: * *p* ≤ 0.05, ** *p* ≤ 0.01. Sample legend: NC—non-stimulated control, LPS—lipopolysaccharide, ManA—manumycin A, ManB – manumycin B, Col—colabomycin E, Asu—asukamycin.

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
