# Peer review of "Inhibition of Pro-Inflammatory Cytokines by Metabolites of Streptomycetes—A Potential Alternative to Current Anti-Inflammatory Drugs?"

_microorganisms, 2020, doi:10.3390/microorganisms8050621_

Round 1

Reviewer 1 Report

In the manuscript: “Inhibition of pro-inflammatory cytokines by metabolites of streptomycetes - a potential alternative to current anti-inflammatory drugs?” by Hrdy et al., authors evaluated the anti-inflammatory effects of small secondary metabolites produced by streptomycetes, manumycin A. The anti-inflammatory effect of commercially available manumycin A was compared to three related compounds: manumycin B, asukamycin and colabomycin E. Time and dose-dependent anti-inflammatory effect of compounds was tested on LPS-stimulated human monocytic leukemia cells THP-1. IL-1b and TNFa were measured by ELISA or RT-PCR. The viability of cells was assessed by Trypan blue staining.

Generally, bioactive secondary metabolites produced by streptomycetes have significant potential to be used in medicine, thus, the submitted study is clinically relevant. However, there are several issues to be clarified before publishing.

Abstract:

Less general information but more and detailed information about used substances/compounds would be useful.

Introduction:

  • As in the Abstract, more information about the source of the compounds, reasons for selection of specific compounds, reasons why authors measured only IL-1-beta and TNF-alpha and not other cytokines.
  • Certain symbols, such as “alpha” “kappa” are missing in the text (please, check throughout the whole manuscript)
  • Line 61: I suggest to complete the sentence ….agonists of the innate immune response
  • If not a general knowledge, please, add cited literature e.g. Line 47, 48, 59, 72…
  • Line 92: do you mean assess instead of assay?
  • Please, check the Fig 1 manA: Is one double bond missing?

Materials and Methods:

Line 125: streptromycin?

Results

  • Did you measure the content of LPS in the compounds?
  • Please, make clear in Figures, that tested compounds were co-incubated with LPS
  • Did you perform an experiment where cells were stimulated only with the compounds? Did you co-culture the cells and compounds with different TLR ligand, such as Pam3cys?
  • Is the anti-inflammatory effect TLR-4-dependent?
  • How many times was the experiment repeated?

Author Response

Response to reviewers´s comments and suggestions

We would like to thank you, the editorial committee and sincerely thank the reviewers for their positive comments and for giving us the opportunity to submit a revised version of our manuscript to the journal Microorganisms. We notably agree with the fact that certain symbols from Greek alphabet has been changed. For that reason, we have checked the whole manuscript and corrected them. We revised our manuscript to address all concerns of the reviewers. Please find enclosed the rebuttal letter with point-to-point responses (in blue) to the reviewer’s comments.

Reviewer 1

In the manuscript: “Inhibition of pro-inflammatory cytokines by metabolites of streptomycetes - a potential alternative to current anti-inflammatory drugs?” by Hrdy et al., authors evaluated the anti-inflammatory effects of small secondary metabolites produced by streptomycetes, manumycin A. The anti-inflammatory effect of commercially available manumycin A was compared to three related compounds: manumycin B, asukamycin and colabomycin E. Time and dose-dependent anti-inflammatory effect of compounds was tested on LPS-stimulated human monocytic leukemia cells THP-1. IL-1b and TNFa were measured by ELISA or RT-PCR. The viability of cells was assessed by Trypan blue staining.

Generally, bioactive secondary metabolites produced by streptomycetes have significant potential to be used in medicine, thus, the submitted study is clinically relevant. However, there are several issues to be clarified before publishing.

We sincerely thank the reviewer for these positive remarks and for appreciating our work. As requested, we have taken into account the comments in order to clarify some parts of the manuscript and also to modify some statements we made.

Abstract:

Less general information but more and detailed information about used substances/compounds would be useful.

The majority information in the literature describes manumycin A because it is commercially available for a long time. Concrete information about manumycin A has been added as follows „ Anti-cancer effect of manumycin A has been previously well documented, micromolar concentrations suppress inflammatory response.“ To keep the word limit 200 words, some general part in the Abstract has been removed: Some macrolide (e.g. azithromycin) and cyclic oligopeptide (daptomycin) antibiotics have been proven to possess anti-inflammatory activity that can positively influence inflammatory diseases.

Introduction:

  • As in the Abstract, more information about the source of the compounds, reasons for selection of specific compounds, reasons why authors measured only IL-1-beta and TNF-alpha and not other cytokines.

Thank you for these comments. The source of the compounds are streptomycete strains (Asukamycin, Colabomycin) and manumycins A and B have been purchased. Information are provided in Material and Methods section. We have specified strains on line 113 (to be sure that they are streptomycetes).

In addition to that, detailed information about manumycin-type metabolites has been introduced in the manuscript as follows “The family of manumycin-related compounds is rather large, it includes several tens of related molecules with the highest structural variability found in the upper polyketide chain. Only few of them are produced by the producer strains in the amounts allowing purification of pure compounds in the amounts needed for detailed activity tests, most of them appear as side metabolites, or congeners of the main compound. This was one of the limits in the compound selection. Next, the selected compounds vary in the structure of their upper chains substantially – manumycins A and B have partially saturated, branched upper chains, asukamycin’s upper chain is unsaturated linear with a cyclohexyl attached in the end and colabomycin E has a long, linear, unsaturated chain with an unusual cis isomery of a double bond.”

The reason why we measured pro-inflammatory cytokines IL-1beta and TNF-alpha is stemming from our experience with our model for screening the novel compounds for their anti-inflammatory properties. We have observed that cytokines IL-1beta and TNF-alpha are the most sensitive markers reflecting downregulation of inflammation using THP-1 cell line in our experimental set up. Actually, we have tested other cytokines and factors involved both in pro-inflammatory and anti-inflammatory responses previously (Cecrdlova et al. Immunology Letters, 2016: 169: 8-14). Other pro-inflammatory marker - IL-6 is affected by Streptomycetes´metabolites but it does not reached statistical significances in the most cases. If you find it interesting, we will be happy to provide these results as a supplementary file. Similarly, anti-inflammatory cytokine, IL-10, has been tested but there was only marginal dose dependent effect. In our previous work (Cecrdlova et al. Immunology Letters, 2016: 169: 8-14), different genes were tested but the most evident effect are consistently demonstrated for IL-1beta and TNF-alpha. Therefore, the preliminary data suggested to use these cytokines as markers of anti-inflammatory potential of novel drugs/compounds for the primary screening.

In the revised version of the manuscript, information regarding cytokine selection has been included as follows “Based on our previous observation, cytokines IL-1β and TNF-α are the most sensitive markers reflecting downregulation of inflammation using THP-1 cell line in our experimental setup [13]. Therefore, the capacity of manumycin-type metabolites to suppress both gene expression and secretion of IL-1β and TNF-α has been tested.”

  • Certain symbols, such as “alpha” “kappa” are missing in the text (please, check throughout the whole manuscript)

Thank you for notifying us. We are really sorry for misspelling letters from Greek alphabet. We have check the manuscript thoroughly and corrected all the wrong symbols.

  • Line 61: I suggest to complete the sentence ….agonists of the innate immune response

Thank you for this suggestion. The sentence is now clearer.

  • If not a general knowledge, please, add cited literature e.g. Line 47, 48, 59, 72…

Thank you for your remark. We have added citation to the mode of action of Tofacitinib, Enaminone E121 and TAK-242. FK506 is widely used in transplantation medicine. We believe that citation is not needed but if you prefer to add some reference, we will be happy to provide it. Similarly, the capacity of polyketides to affect enzymatic activity is generally known.

  • Line 92: do you mean assess instead of assay?

Thank you for this correction. You are right, assess suits much better in this sentence.

  • Please, check the Fig 1 manA: Is one double bond missing?

We apologize for this mistake. New figure with the double bond is provided in the revised version of the manuscript. Thank you for your notice.

Materials and Methods:

Line 125: streptromycin?

We apologize for this typo. Of course, it is streptomycin. This has been corrected in the revised version of the manuscript.

Results

  • Did you measure the content of LPS in the compounds?

Thank you for this suggestion. LPS is known as strong activator of macrophages (THP-1 cell line), therefore the presence/contamination of the compounds by LPS could induce pro-inflammatory responses. We did not measure the presence of LPS in the compounds. We have several reasons to believe that the compounds do not contain LPS: A) Streptomycetes do not contain LPS. B) They may, theoretically, appear as the contamination of the cultivation media components or water in the fermentation experiment, however, both LPS and lipid A are quite polar, they will not extract together with manumycins by non-polar organic solvents and they will not contaminate the fractions containing the compounds during the chromatographic purification steps as they do not share the molecule size and polarity with manumycins. C) Theoretically, the isolations may contain LTA (lipoteichoic acids), as streptomycetes contain these, however, similar to LPS, they should not co-purify with the manumycin-type compounds due to the different physical-chemical properties. D) The purified compounds do not cause activation of macrophages, but rather inhibition as we have described in the current manuscript.

  • Please, make clear in Figures, that tested compounds were co-incubated with LPS

We have modified Figures to be clearer that compounds were co-incubated with LPS. Please see Figures 2-4.

  • Did you perform an experiment where cells were stimulated only with the compounds? Did you co-culture the cells and compounds with different TLR ligand, such as Pam3cys?

Thank you for this question. Indeed, we have performed experiments where the impact of manumycin-type metabolites was tested on THP-1 cells without LPS stimulation. As the cells were not directed into pro-inflammatory immune response by LPS stimulation, the impact of manumycin-type metabolites on downregulation of pro-inflammatory markers (e.g. IL-1beta, TNF-alpha) was only marginal. To see the effect of manumycin-type metabolites on suppression of pro-inflammatory responses, the cells THP-1 have to be stimulated/primed into pro-inflammatory responses. We agree, it would be interesting to stimulate THP-1 cells by Pam3cys. We have not done this but we stimulated THP-1 cells by recombinant TNF-alpha and we observed a similar effect. Actually, this has been reported by (Cecrdlova et al. Immunology Letters, 2016: 169: 8-14). In the current experimental set up, the effect of LPS stimulation of THP-1 cells was superior to TNF-alpha stimulation.

  • Is the anti-inflammatory effect TLR-4-dependent?

Thank you for this excellent question. Honestly, we do not know but our idea is that the anti-inflammatory effect of our compounds is not mediated via TLR4 signalling. It has been published that manumycin A targets IKKbeta (https://www.jbc.org/content/281/5/2551.full.pdf). The epoxide group of manumycin is responsible for homodimerisation of IKKbeta, therefore, preventing formation of IKK. NF-kappaB cannot be active due to the lack of active IKK. All manumycin-type metabolites tested in this paper possess this epoxide group, therefore we are expecting that they will work in the same manner as manumycin A in the case of inhibition of IKK formation.

How many times was the experiment repeated?

Results are from three independent experiments. We have added this information in Statistic section as follows: Results are expressed as a mean with standard error mean from three independent experiments.

Reviewer 2 Report

Dear author/s,

I recommend your paper to be published with minor changes as follows:

row 13: I recommend to include "among others" in between "includes" and "application".

row 15: "...treatment is quite often inefficient..." is very strong statement, please, change it.

row 29: Change "manumycine" to "manumycin".

rows 43, 77, 78, 87, 139, 147, 192, 193, 208, 308, 309 : Greek letters, such as beta and alpha in cytokines,

kappa in NF-kB, micro in uM etc., are missing.

row 68: Add more references to be it in concordance with "...in other works....".

row 82: Add "." to the end of sentence.

row 84: Change "anticancer" to "anti-cancer".  Delete "-see following references for examples".

row 115: Change "(Cecrdlova et al., 2016)" to "⌈?⌋" as other references in the paper.

rows 132, 136: Change "(Hrdý et al., 2010)" to "⌈?⌋" as other references in the paper.

rows 179, 206, 219, 230, 262, 266: Change "Fig." to "Figure".

row 185: Delete ")" after 0.0098.

row 189: Change "of expression of TNF-α" to "of TNF-α expression".

row 197: Change "Other concentration tested," to "Other tested concentrations".

row 198: Delete "." after "cells".

row 201: Chance "concentration" to "concentrations".

row 203, 215: Delete "used".

row 252: Change "Fig 3B" to "Figure 4A".

row 264: Add "tested" between "highest" and "concentration".

row 265: Change "of release of IL-1β and TNF-α" to "of IL-1β and TNF-α release".

row 295: Change ": The" to ": the".

row 326: Add "drugs" or "medicines" after "anti-cancer".

Author Response

Response to reviewers´s comments and suggestions

We would like to thank you, the editorial committee and sincerely thank the reviewers for their positive comments and for giving us the opportunity to submit a revised version of our manuscript to the journal Microorganisms. We notably agree with the fact that certain symbols from Greek alphabet has been changed. For that reason, we have checked the whole manuscript and corrected them. We revised our manuscript to address all concerns of the reviewers. Please find enclosed the rebuttal letter with point-to-point responses (in blue) to the reviewer’s comments.

Reviewer 2

Dear author/s,

I recommend your paper to be published with minor changes as follows:

We sincerely thank the reviewer for this positive comment and for suggestions how to improve our manuscript.

row 13: I recommend to include "among others" in between "includes" and "application".

Thank you for this suggestion. We have included this in the abstract.

row 15: "...treatment is quite often inefficient..." is very strong statement, please, change it.

Thank you for this notice. We have replaced „is quite often inefficient“ by „can be inefficient“

row 29: Change "manumycine" to "manumycin".

We are apologizing for this typo. It has been corrected to manumycin.

rows 43, 77, 78, 87, 139, 147, 192, 193, 208, 308, 309 : Greek letters, such as beta and alpha in cytokines, kappa in NF-kB, micro in uM etc., are missing.

Thank you for notifying us. We have tried to correct it. We are really apologizing for this misspelling Greek letters.

row 68: Add more references to be it in concordance with "...in other works....".

Thank you for this suggestion. Actually, our unpublished observation has been published and we have added additional three references in the revised version of the manuscript. “Our pilot experiments show that immunomodulatory features can often be identified in secondary metabolites extracts of many human-associated streptomycetes [15] and similar reports scarcely appear in other works, too [16-19].“

Links to the papers:

https://www.jstage.jst.go.jp/article/antibiotics1968/46/9/46_9_1334/_pdf/-char/en

row 82: Add "." to the end of sentence.

The point has been added at the end of the sentence. We are sorry for omitting the point.

row 84: Change "anticancer" to "anti-cancer".  Delete "-see following references for examples".

Thank you for this suggestion. We have included hyphen in the word „anti-cancer“. The redundant part of the sentence has been deleted.

row 115: Change "(Cecrdlova et al., 2016)" to "⌈?⌋" as other references in the paper.

Thank you for this notice. The reference has been unified and the text „"(Cecrdlova et al., 2016)" has been replaced by the number 24 in the brackets according to the journal citation standard.

rows 132, 136: Change "(Hrdý et al., 2010)" to "⌈?⌋" as other references in the paper.

Similarly, the citing reference has been replaced by the number – 35. Thank you for this remark.

rows 179, 206, 219, 230, 262, 266: Change "Fig." to "Figure".

Thank you for your recommendation. We have replaced abbreviation Fig. with the full length word.

row 185: Delete ")" after 0.0098.

The part of the bracket has been deleted.

row 189: Change "of expression of TNF-α" to "of TNF-α expression".

Thank you for this suggestion. This has made the sentence clearer.

row 197: Change "Other concentration tested," to "Other tested concentrations".

Thank you for this recommendation. The text of the manuscript has been modified accordingly.

row 198: Delete "." after "cells".

The point has been removed.

row 201: Chance "concentration" to "concentrations".

You are right. The plural is the correct form. The text of the manuscript has been corrected.

row 203, 215: Delete "used".

Thank you. This changes simplified the text.

row 252: Change "Fig 3B" to "Figure 4A".

We apologize for the incorrect number of the figure. This has been corrected in the manuscript.

row 264: Add "tested" between "highest" and "concentration".

Thank you. We have included „tested“ in the test of the manuscript according to your recommendation.

row 265: Change "of release of IL-1β and TNF-α" to "of IL-1β and TNF-α release".

Thank you for this suggestion. The text in the manuscript is now clearer.

row 295: Change ": The" to ": the".

The capital letter „T“ has been changed to lower case.

row 326: Add "drugs" or "medicines" after "anti-cancer".

Thank you for this recommendation. We have added the word „drugs“.
